# Experiences and Perspectives of Children and Young People Living with Childhood-Onset Systemic Lupus Erythematosus—An Integrative Review

**DOI:** 10.3390/children10061006

**Published:** 2023-06-02

**Authors:** Julie Blamires, Mandie Foster, Sara Napier, Annette Dickinson

**Affiliations:** 1School of Clinical Sciences, Auckland University of Technology, Auckland 0627, New Zealand; mandie.foster@aut.ac.nz (M.F.);; 2School of Nursing and Midwifery, Edith Cowan University, Perth 6027, Australia

**Keywords:** adolescent, anxiety, depression, diagnosis, lupus erythematosus, systemic, medication adherence, youth, social impact, quality of life

## Abstract

Childhood-onset systemic lupus erythematosus (cSLE) impacts the daily life of children and young people. This study aimed to describe the experiences and perspectives of children and young people living with cSLE. An integrative review guided by Whittemore and Knafl was conducted. Extant empirical research published in peer-reviewed journals from 2000 to 2021 on children’s self-reported experiences living with cSLE was identified from Scopus, CINAHL, Medline via PubMed, and PsycINFO via Ovid databases. Nineteen studies involving over 1400 participants were included. Four themes and fourteen sub-themes were identified: (1) challenging symptoms (disruptions to life and altered self, severity, fatigue, depression, and anxiety), (2) medicines and side effects (dreaded steroids, conflicting feelings, and medication adherence), (3) complicated life (school sports and social, giving things up, lack of understanding, and quality of life) and (4) ways of coping (family and friends, relationships with health providers, and maintaining positivity). While cSLE shares many similarities with adult-onset SLE, awareness of differences in experiences and perceptions of children and young people is crucial. The significant psychological and social impact of the disease and its treatments necessitates a comprehensive, holistic approach to managing cSLE that considers the unique needs of youth.

## 1. Introduction

Childhood-onset systemic lupus erythematosus (cSLE) is a rare multi-system autoimmune disease that is defined by disease onset before the age of 18 years [1]. It affects approximately 15–20% of SLE patients and has a variable presentation and clinical course [2,3]. Incidences of cSLE are reported as between 0.36 and 2.5 per 100,000 children, with prevalence ranging from 1.89 to 34.1 per 100,000 [4,5,6,7]. Similar to adult onset (aSLE), cSLE is clinically complex, heterogeneous, and characterized by periods of flare and remission [8]; however, cSLE is much more aggressive than aSLE with higher disease activity, more severe organ manifestations, and subsequently increased medication burden [9,10,11]. Childhood-onset systemic lupus erythematosus most commonly affects the skin (including malar and discoid rash), muscles, and joints; however, involvement of major organs is common, with renal inflammation occurring most frequently [2,12]. Other non-specific symptoms of fatigue, pain, weight loss, and fever are frequent and result in significant morbidity [10].

Research in aSLE has examined experiences and outcomes through the analysis of barriers and facilitators, exploration of beliefs, illness perception, and treatment adherence [13,14,15,16]. In addition, studies describe how disease activity, anxiety, social support, sleep quality, and fatigue negatively impact quality of life (QOL) in aSLE [17,18]. In cSLE, much of the focus has been on treatment targets [8], adherence [19,20], health literacy and/or knowledge deficits [21], and the challenges of disease management [22]. Previous studies in other pediatric rheumatological conditions have demonstrated significant disruptions to the trajectory of young people’s lives, impacts on relationships, effects on self-esteem, and the creation of more challenges for children and young people (CYP) in day-to-day life [23,24]. In addition, mental health problems, particularly anxiety and depression, are more common in CYP with pediatric rheumatologic diseases compared to healthy peers [25].

Children and young people with cSLE develop the disease at a young age, have a longer disease course, and therefore have more time to develop long-term complications and multi-system co-morbidities [9,26]. In addition, the physical and psychosocial changes that occur with maturity, the increased likelihood of requiring more intensive drug therapy, and the necessity to have parents and caregivers involved in management contribute to the complexity and equate to a clinical course that is unpredictable and challenging for CYP and their family [27]. Given that CYPs’ needs are different from adults [28] and that physical, psychosocial, and emotional development impacts and influences the way they experience and cope with cSLE [11], it is imperative to view and understand this unique perspective from CYPs’ self-reported experiences to guide practice, research, and theory. Therefore, the aim of this review was to critically appraise and synthesize research that examines CYP’s experience and perspectives of living with cSLE.

## 2. Materials and Methods

An integrative review design enabled the inclusion of studies from a broad range of methodologies [29]. A comprehensive review of both quantitative and qualitative evidence was considered important in addressing the study’s objectives and identifying new insights and research opportunities. Whittemore and Knafl’s [30] integrative review framework was used, allowing for a rigorous review and synthesis of extant empirical research on the review topic.

### 2.1. Problem Identification and Search Strategy

An initial scoping review of databases was undertaken to refine the research question and focus the search strategy. A systematic search of databases (Scopus, CINAHL, Medline via PubMed, and PsycINFO via Ovid) was conducted in August 2021. A combination of search terms was utilized for population, exposure, and outcomes/themes; see Table 1. SLE Search terms.

Boolean operators and truncation symbols enhanced search results, while proximity searches increased specificity. Keywords included text words, abbreviations, and truncated text words (Appendix A. Search Architecture). Limiters to the search included publication dates from 2000 to 2021 and empirical research published in peer-reviewed journals in the English language. Additional records were retrieved from manually reviewing the table of contents of relevant journals and reference lists of papers on the topic.

### 2.2. Eligibility Criteria

The inclusion criteria were: (1) original qualitative, quantitative, or mixed method research studies reporting on CYP’s experiences and perceptions of cSLE, including pain, lifestyle, activity levels, decision making, medication, quality of life, self-management, social impact, and patient-reported outcome measures; (2) participant’s aged 0–24 (based on children 0–18 and young people 10–24 as defined by WHO and the United Nations) [31,32]; (3) peer-reviewed journals in the English language: (4) published between 2000 and 2021. Abstracts, conference reports, editorial letters, the literature not formally published, and studies where the CYP’s voices could not be extrapolated from a parent–child dyad study were excluded.

### 2.3. Quality Appraisal

All included articles were appraised for quality using the Mixed Methods Appraisal Tool (MMAT) [33]. The MMAT is a critical appraisal tool that allows studies using various designs to be measured by similar quality appraisals, assessing studies on their generic research approach, individual components, and/or mixed methods approach [34]. Three authors independently assessed the quality of the included articles using the appropriate criteria from the MMAT [34]. (Appendix A Mixed Methods Appraisal Tool Combined Results). If there were any disagreements between scores, these were then discussed with the research team until a consensus was agreed upon. No manuscripts were excluded based on a low critical appraisal score.

### 2.4. Data Management and Extraction

A data extraction form was developed (see Appendix A, data extract form) according to the Joanna Briggs Institute [35] and was categorized into the following subsections: study description, aim, design, sample/participants, location, data/quotes, and analysis against questions/outcomes of the paper. Initially, two authors extracted results from the included papers, and then three authors conducted a secondary review of the extracted data [35]. Results were recorded and exported to EndNote 20.4. (Bld 16297).

### 2.5. Data Synthesis and Analysis

Following Whittemore and Knafl’s [30] framework, all data from the nineteen articles were extracted and summarised in a data extraction table. This enabled the authors to understand what was known about CYP’s self-reported experience of living with cSLE and set the stage for inductive thematic data analysis [30]. Two authors independently identified initial codes, and all authors discussed the patterns in the data, grouping the codes into themes that encompassed the data.

## 3. Results

A total of 1636 articles were screened through database searches. Of these, 1617 were excluded because they did not meet the inclusion criteria (Figure 1). Nineteen studies were included in the review that involved more than 1400 CYP ranging in age from eight to twenty-four years of age [36,37,38,39,40,41,42,43,44,45,46,47,48,49,50,51,52,53,54]; see Table 2 for Characteristics of Included Studies. Eleven studies used a quantitative design, seven a qualitative design, and one study followed a mixed-methods approach, as represented in Table 2. The studies were conducted across nine countries: the United States of America (11), Brazil (1), Columbia (1), China (1), Italy (1), the United Kingdom (1), Singapore (1), Australia (1), and Turkey (1). The included studies had MMAT scores ranging from four to five, with the Harry et al. (2019) study receiving the lowest score of two (Appendix A Mixed Methods Appraisal Tool combined results).

We identified four themes and fourteen sub-themes: (1) challenging symptoms (disruptions to life and altered self, severity, fatigue, depression, and anxiety), (2) medicines and side effects (dreaded steroids, conflicting feelings, and medication adherence), (3) complicated life (school sports and social, giving things up, lack of understanding, and quality of life) and (4) ways of coping (family and friends, relationships with health providers, and maintaining positivity). See Figure 2. Key themes representing young people’s experience of living with cSLE. Selected quotations to illustrate each sub-theme and the articles where the sub-themes were evident are provided in Table 3. Illustrative Quotes.

## 4. Challenging Symptoms

**Disruption to Life and Altered Self.** The journey to a diagnosis was the first disruption experienced in the lives of CYP and resulted in feelings of resentment and stress [39,44,54]. Several studies reported that the first symptoms experienced at diagnosis were particularly challenging, as was the period of waiting for a diagnosis and trying to find appropriate treatment [37,49,52,54]. Participants described how at the time of diagnosis, they experienced self-doubt and questioned whether they possibly imagined symptoms [39,43,44,54]. Once the diagnosis was made, CYP found their sense of normalcy disrupted, and although they had confirmation of their symptoms, sharing the diagnosis with others was difficult, with some CYP choosing to keep their condition hidden [39] or preferring not to be defined by it [54].

Skin rashes, joint pain, muscle aches or weakness, loss of mobility, hair loss, and fatigue severely impacted CYPs’ self-perception [44,45,50,52] as they no longer saw themselves as young and healthy but instead sick and incapacitated. The severe emotional impact of cSLE resulted in feelings of grief and resentment towards the disease and a strong wish that cSLE had never entered their lives [50]. Two studies highlighted how symptoms severely influenced how CYP felt about their appearance, including making them feel unattractive [45], giving them a heightened sense of self-consciousness, and contributing to poor self-image [54]. Some CYP experienced being teased, described as a ‘freak’, or told that they looked ‘weird’ in school, which emphasized the feeling that they were different and not similar to their peers. This notion of being different was evident across other studies where CYP were acutely aware that the symptoms of cSLE imposed limitations on their ability to be normal [44,45,52,54].

**Severity.** The symptoms experienced by CYP were an individual and subjective experience [44,45,52,54]. CYP understood that cSLE had both minor and major symptoms, and although joint pain, muscle aches, weakness, and rashes were considered common, there was variability between ‘levels of disease’ and to what degree individuals were impacted [52,54]. Living with cSLE was therefore described as a very individual experience, with some participants ‘feeling lucky’ [54] about how it manifested in them, whereas others described their cSLE as more severe [44].

**Fatigue.** Eight of the nineteen studies addressed fatigue and described it as one of the most burdensome symptoms of cSLE [38,41,42,43,44,46,52,54]. Two studies examined fatigue and its relationship to depression and poor health-related quality of life (HRQoL) [38,42]. Furthermore, fatigue was reported to impact school activities, limiting CYP’s capacity to study and perform well at school [46,49,54]. When asked to comment on the strengths and weaknesses of the health-related quality of life/fatigue (Peds QL-FS) measure [52], CYP identified the need for a separate questionnaire to focus on fatigue as the ‘main issue’ with having cSLE [52].

**Depression and Anxiety:** Five studies reported depressive symptoms, anxiety, and suicidal ideation in CYP living with cSLE [41,42,45,46,47]. Assessment of depression was conducted using the children depression inventory (CDI) [42,45,46,47], the Patient Health Questionnaire-9 (PHQ-9) [41], and the Beck Depression Inventory (BDI-II) [47]. Three of these studies reported higher rates of depression and suicidal ideation among CYP living with cSLE compared to age and sex-matched healthy CYP and found links between depression, medication non-adherence, and appearance concerns [41,45,46]. In contrast, Donnelly et al. [42] and Kohut et al. [47] found near-normal values for both depressive symptoms and anxiety among CYP, noting an association between higher levels of depressive symptoms and reduction in HRQoL. Two studies assessed anxiety using the Screen for Child Anxiety Related Emotional Disorders (SCARED) questionnaire [42,46], where 50% of CYP in the Jones et al. [46] study and 34% of CYP in the Donnelly et al. [42] study reported clinically relevant anxiety levels that impacted on HRQoL.

## 5. Medicines and Side Effects

**Dreaded Steroids.** Medication regimes, especially the use of steroidal drugs, raised strong and opposing feelings among CYP [37,40,43,52,53,54]. CYP understood and appreciated the effectiveness of steroids in treating their cSLE [37,52,54] but also loathed the unpleasant side effects of weight gain, swollen face, skin changes (such as acne, striae, and flushing), and increased appetite [39,50,53]. Although the physical side effects were the most distressing aspect of taking steroids, CYP also experienced anxiety, wakefulness, higher physical depressive symptoms, and lower self-esteem [37,47,54]. Ruperto et al. (2004) [51] postulated poorer HRQoL in CYP with active disease or accumulated damage in the renal, central nervous, and musculoskeletal systems, including impaired self-esteem due to changes in body image from the associated side effects of aggressive corticosteroid and immunosuppressive medications. The significant side effects of steroids heightened the sense of self-consciousness, awkwardness, and awareness of being different [44,50].

**Conflicting Feelings.** Four studies described how cSLE medications posed a paradoxical dilemma for CYP, where being dependent on medications for life, multiple times a day, to maintain their health was described as weird, stressful, and nonsensical [40,52,53,54]. This was especially true when CYP believed their cSLE was too mild to warrant treatment or believed that it would not be a ‘big deal’ if they refrained from taking their medications [43,52,53,54]. Because medications were linked to changes in their face and body, social aspects of their life were further impacted; hence medications were not looked upon favorably [44]. However, CYP also came to appreciate the positive aspect of their medications when they started to feel better, and the right medications finally helped mitigate their symptoms [40,52,54]. CYP felt that sometimes health professionals would state cSLE was the reason for their symptoms when in fact, they believed the medications were the cause which created a situation where CYP felt at odds with the treatment plan [54].

**Medication Adherence.** CYP stated they did not take their prescribed medications because of forgetfulness, the number of pills, the bad taste, and/or concern about side effects [39,41,43,45,53]. Some CYP doubted the need for medications because they deemed they were not sick enough or felt their cSLE ‘wasn’t that bad’ [52,53], whereas others described a lack of perceived improvement in daily symptoms [43] as a barrier to adherence. Other common issues that impacted adherence were the logistics around taking medications, disruptions to normal activities, and sometimes business and/or getting out of the routine [39,43,53]. Two studies stated that adherence was influenced by a perceived lack of transparency from health professionals about the side effects [53,54].

Case et al. (2021) reported that a trusting patient–provider relationship positively impacted medication adherence, especially when CYP perceived healthcare professionals were working in their best interests. Other key factors that contributed to adherence were: recognizing the benefits through the acquisition of knowledge about cSLE; understanding how medication helped them control their disease, including reducing flares; and appreciating that medications enabled them to engage in activities to achieve their aspirations [40,43,53]. In addition, Tan et al. (2021) reported that parents’ attitudes, involvement, and strict monitoring of medication were critical to adherence.

## 6. Complicated Life

**School, Sports, and Social Activities.** The burden of treatment, constant clinic appointments, side effects of medication use, and symptoms all impacted daily life, including school attendance, performance [43,49], and sports and leisure activities [44,54]. CYP described a desire to do well at school but conceded that the unpredictable nature of cSLE resulted in a lack of control over educational pursuits [37,49]. Being able to participate and being included in normal school activities was described as a key way of coping with cSLE in a social context [50].

**Giving Things Up.** The sense of having to ‘give things up’ or being limited in pursuing life’s usual adventures, such as back packing or living overseas, was a source of frustration [44,50,54]. In addition, the prognostic uncertainty of cSLE led CYP to feel worried about their future and ability to achieve long-term goals [43,44,50,54]. This was reflected in their perceptions of limitations regarding educational and career aspirations, anxiety about the worsening or persistence of disease, concerns about a shorter lifespan, and the potential effect on their ability to have a family of their own [43,44,46,50,54].

**Quality of Life.** Six studies reported on how cSLE impacted HRQoL [36,38,42,46,48,51]. These studies used both generic tools such as the Pediatric Quality of Life Inventory Generic Core scale (PedsQL-GC) and the Child Health Questionnaire (CHQ) [36,37,42,46,48] as well as rheumatology-specific tools such as the Rheumatology Module (PedsQL-RM) [42,46,48,51] and Systemic Lupus Erythematosus Disease Activity Index (SLEDAI) and British Isles Lupus Activity Group index (BILAG) tools [42,48,51]. Overall, when compared to healthy CYP, those with cSLE had poorer HRQoL across both physical and psychosocial domains [38,42,46,48,51] with the exception of Uzuner et al. (2017) [36], who found no difference in their study. When looking at specific symptoms and/or features of cSLE, the studies reported on a variety of factors that negatively impacted HRQoL, including the presence of musculoskeletal and general symptoms such as pain, mood, fatigue, and anxiety [38,42,46,48]; organ-specific disease damage particularly renal, central nervous, and musculoskeletal systems [38,46,48]; gender; and the use of cyclophosphamide and/rituximab [48].

**Lack of Understanding.** Coming to understand what cSLE is and how they came to have it resulted in CYP making associations and interpretations based on: what they were told by health professionals [43,44], explanations from parents [37,44], and information sources from the internet [54]. Sometimes these information sources were deemed to be too ‘sciencey’ [43], whereas others claimed they wanted to know everything they possibly could [54]. Children and young people also felt adults did not always provide the full picture, whereas it was important for CYP to know that cSLE was not going to go away [39,54].

Misunderstanding the nature of cSLE, the aetiology, and how it impacted CYP’s lives were commonly encountered [44,54]. There was a sense among CYP that family, friends, teachers, and even health professionals did not appreciate or understand the difficulty of living with cSLE, and consequently, empathy was lacking [39,43,44]. Misinformation created negative feelings, especially when it led to peers frantically wiping down surfaces for fear they would ‘catch’ cSLE [43,50]. This lack of understanding was deemed particularly important in relation to schools where CYP felt that education from health professionals to teachers might help with some of the challenges they faced [43]. More public exposure and the need for disease-specific information about cSLE were considered highly important [43,50,54].

## 7. Ways of Coping

**Family and Friends.** CYP described how talking to others with cSLE, either in person or online, was valuable for gaining a sense of belonging, being understood, and enhancing overall wellbeing [37,39,54]. Family members were deemed invaluable for providing emotional and practical support, access to treatment, encouraging medication adherence, and providing information about cSLE [37,39,43,44,50,53,54]. Several studies addressed the role that friends/peers played [39,53,54]. Relationships with peers provided invaluable emotional support that enabled CYP to cope and manage their life with cSLE [39,50]. Being included and encouraged to join normal activities [50] or being ‘cheered up’ about changes in appearance due to medication side effects [53,54] helped to combat feelings of being different from peers.

**Relationships with Health Providers.** Connectedness and collaborative relationships with health providers were key to coping with cSLE. Having trust and confidence in their health care provider enabled participants to manage cSLE and contributed to treatment adherence [39,43,53,54]. Studies highlighted how CYP highly valued health care professionals who were accommodating and took into consideration individual needs when developing treatment plans [39,43,53]. Connections and relationships with healthcare professionals were noted to be particularly important during the transition from pediatric to adult services [39,43,54]. The transition came with tension and unanticipated challenges for CYP; thus, willingness to transition was strongly influenced by the bonds they had with their first healthcare provider, which was usually the CYP’s rheumatologist [39,54].

**Maintaining Positivity.** Despite the many challenges of living with cSLE, CYP strived to maintain a positive attitude by viewing cSLE as not merely a challenge to overcome but also a means of developing self-confidence, empathy, resilience, and self-care abilities [39,44,50,53,54]. Thus, important life skills such as coping, empathy, and resiliency were acquired [39]. Further, CYP gained self-management skills, such as learning about their body and how to take care of it [44]. Others described how being focused and positive about learning to self-manage their medications prevented their family members from worrying [53]. In addition, some CYP experienced a recalibration of mindset and considered themselves ‘lucky’ to have cSLE rather than something they deemed more serious, such as cancer [39,44,54]. Finding a way to focus on the positive aspects of life, being grateful for the little things, and re-evaluating priorities were described as ways of coping [44,49,53,54].

## 8. Discussion

It is evident within this review that CYP living with cSLE experience a wide range of visible and invisible symptoms, including pain, fatigue, rashes, depression, anxiety, and altered body image that disrupts their normal psychosocial, emotional, physical, and developmental growth, and places them at higher risk of bullying, and adverse mental health outcomes than their peers. These findings are similar to the experiences of CYP with other rheumatological diseases where fatigue, anxiety, and depression have been shown to have a significant impact not only on HRQoL but day-to-day emotional and psychosocial wellbeing [23,55,56,57]. The impact and severity of cSLE were highlighted by this review as highly unique and, in the wider literature, influenced by individual demographic and illness characteristics, with a level of impact being strongly linked to disease activity and organ damage [2,58,59]. In addition, gender, age of diagnosis, type of immunosuppressive medication, number of flares, and ethnicity are factors that influence CYP’s likelihood of increased mortality [58,59,60].

Children and young people stated that the first step into their cSLE journey was waiting for a diagnosis, having their symptoms confirmed, and commencing treatment which triggered emotional turmoil, feelings of relief, and significant frustration. The difficulty in making an initial diagnosis of cSLE in CYP has been attributed to its multi-system involvement, atypical symptoms, variability in clinical presentation [2,58,61], as well as other factors, including demographic, clinical, and socioeconomic characteristics [62]. Given the importance of prompt diagnosis to prevent disease-related morbidity, Hussain et al. (2022) recommended the need for screening guidelines and policies to improve healthcare outcomes for CYP.

Medications played an important part in the management of cSLE and CYP’s experiences of living with it, including their self-perception, self-esteem, and relationships with healthcare providers. Children and young people described feeling anxious and worried about the side effects of medications particularly the Cushingoid features that accompanied the use of glucocorticoids. The changes to their body, mood, and sense of self related to steroid treatment represented a confronting and psychologically distressing aspect of cSLE. Glucocorticosteroids remain a foundational treatment in SLE [63]; however, it is increasingly recognized that the dose should be kept as low as possible in order to minimize both short and long-term side effects [64]. In addition, there is a recognized need to evaluate the side effects and benefits of steroid use from CYP’s perspective [65]. Given that most CYP in this review were adolescents, concerns about physical appearance and disturbance in body image were not surprising. Children and young people are extremely sensitive to and focused on changes to their appearance and the opinions of peers [66]. Alterations to their self and body image and to the psychological distress that may accompany changes potentially place them at increased risk of depression when compared to their healthy peers [67,68] and those with adult-onset SLE [69]. This points to the need for routine psychosocial screening of CYP with cSLE by health professionals to ensure at-risk and psychologically compromised CYP are identified and supported [24,70]. In addition, there is a need for increased awareness in practice and further research to identify strategies for improving early intervention for depression and anxiety in CYP with cSLE [57].

The conflicting feelings surrounding medication adherence were related to the burden of having to take medications every day, the disruption to normal life, and the potential risk of side effects; however, positive effects were experienced when medication improved CYP’s condition. Treatment adherence was influenced by having a trusting relationship with healthcare providers, feeling involved with decision-making about treatment, and developing personal knowledge and understanding about cSLE. These findings are consistent with adult studies and those related to other rheumatological conditions that describe the importance of resilience [71], the healthcare provider–patient relationship [72], and patient empowerment [73] as positively impacting medication adherence. Increasing CYPs’ knowledge about cSLE and medications and addressing concerns about side effects have been described as important ways to improve communication between CYP and healthcare professionals and improve adherence [16,74,75]. For CYP, this may require consideration of different approaches and broader access to educational material, such as online or web-based interventions and the use of social media, shown to enhance adherence and support young people in managing their cSLE medications [76,77]. Many CYP living with cSLE will never achieve an inactive disease state; therefore, medication adherence is a particularly important aspect of care, especially when using high immunosuppressive treatments [58,60].

Disruptions and interruptions to normal activities due to the burden of treatment, clinic appointments, side effects of medications, and feeling restricted in their ability to achieve future goals and aspirations created a complicated life for CYP living with cSLE. A correlation between neuropsychiatric symptoms, cognitive disorders, and poor academic performance in maths, learning, attention, information processing, memory, and completing school or college degrees has been reported in the literature as a significant factor for CYP with cSLE being unable to reach academic outcomes and aspirations [49,78,79]. Many CYP in this review spoke about how they felt school, family, and friends did not fully understand or appreciate the difficulties they faced living with cSLE and expressed concern about the lack of public knowledge regarding their disease. This was echoed in a study from the United Kingdom, where CYP felt that greater public awareness and wider communication about rheumatic conditions were needed, believing this would enable them to manage their daily challenges more easily [80]. Children and young people also recommended that research should focus on the lived experience of CYP to gain a greater understanding of the psychosocial impact of the illness, with CYP being actively consulted, included, and viewed as expert informants (64).

Social support systems, including peers, family, and cSLE support groups, were found to be necessary and important for coping with cSLE by CYP in this review. Organized events, such as camps for CYP with cSLE, were beneficial in providing social support through shared experiences and helping young people to normalize their life [81,82]. Camps for CYP offer opportunities for peer acceptance and social inclusion, and these benefits have been similarly described in studies in other long-term conditions such as diabetes [83], celiac disease [84], bleeding disorders [85], and epilepsy [86]. It is reported in the literature that CYP with cSLE has shown high coping scores for emotion-focused coping strategies (0.6 ± 0.2) and socially supported coping strategies (0.5 ± 0.2) linked to high happiness scores on the Brief Coping Orientation to Problems Experienced (COPE) questionnaire and the Subjective Happiness Scale (SHS) [87]. Remaining grateful and building on one’s self-confidence, resiliency, and self-care abilities in managing medications, treatment, and symptoms were some of the problem-focused coping strategies identified in this review. Similarly, CYP with cSLE had high self-reported coping scores for problem-focused coping (0.7 ± 0,2) associated with high levels of happiness (r = 0.564, *p* = 0.002) [87]. Despite these positive findings, CYP living with cSLE require individualized, multi-dimensional interventions and support to develop coping strategies to enhance their ability to respond to the multiple challenges imposed by cSLE. This includes the skills needed to effectively transition from CYP services to adult care [82,87,88].

This review highlighted that a multidisciplinary approach is needed that includes routine assessment of both physical and psychosocial symptoms as an essential approach to care, given that early intervention for depression has the potential to improve outcomes. Ongoing psychosocial support, health education, adherence interventions, and personalized treatment plans are necessary components of holistic care for CYP living with cSLE. In addition, there needs to be greater awareness and education to the public and general practitioners about the nature of cSLE, the early symptoms, complications, and treatment strategies. Future health care and research should focus on psychosocial and developmental wellbeing, resiliency, and practical strategies for managing and living with cSLE, fatigue, interventions to promote peer support networks, and medication adherence. Future research needs to be inclusive of CYPs’ voice, including the voices of younger children, and conducted in collaboration with service providers to ensure the best outcomes for CYP living with cSLE.

Limitations. This study, as with all integrative reviews, is limited by search terms, selected databases, and applied search strategy methods. In addition, this review is limited by including only empirical peer-reviewed studies published in English. Most of the instruments were well described, and reliability and validity were reported; however, due to the range and variability of instruments used, a meta-analysis of results was not possible. Not all studies described participants’ SLEDAI or SDI; therefore, it is not possible to draw comparisons between CYP’s experiences in terms of disease activity scores. Finally, CYP living with cSLE were not consulted on the results of this integrative review, given that the project commenced during the COVID-19 pandemic. Despite these limitations, the procedure undertaken by the authors in this review was rigorous and followed a systematic process.

## 9. Conclusions

While cSLE shares many similarities with adult-onset SLE, there are essential differences in terms of morbidity, impact of symptoms, and medication side effects that disrupt CYP’s day-to-day life. An awareness of the differences in experiences and perceptions of CYP is crucial. The significant psychological and social impact of cSLE and its treatment necessitates a comprehensive, holistic approach to managing care that considers the unique needs of CYP. Studies describe fatigue as the most common symptom among adults with SLE [89,90] and highlight the significant impact of depression and anxiety on quality and daily life [91,92]. There is an opportunity for children and young people to identify these symptoms earlier and potentially prevent these impacts as children grow into adulthood. The review highlights significant anxiety, fatigue, and depression among CYP due to cSLE and points to the need for further research and exploration to understand CYPs’ experiences and their support needs to more fully direct practice, research, policy, and knowledge.

## Figures and Tables

**Figure 1 children-10-01006-f001:**
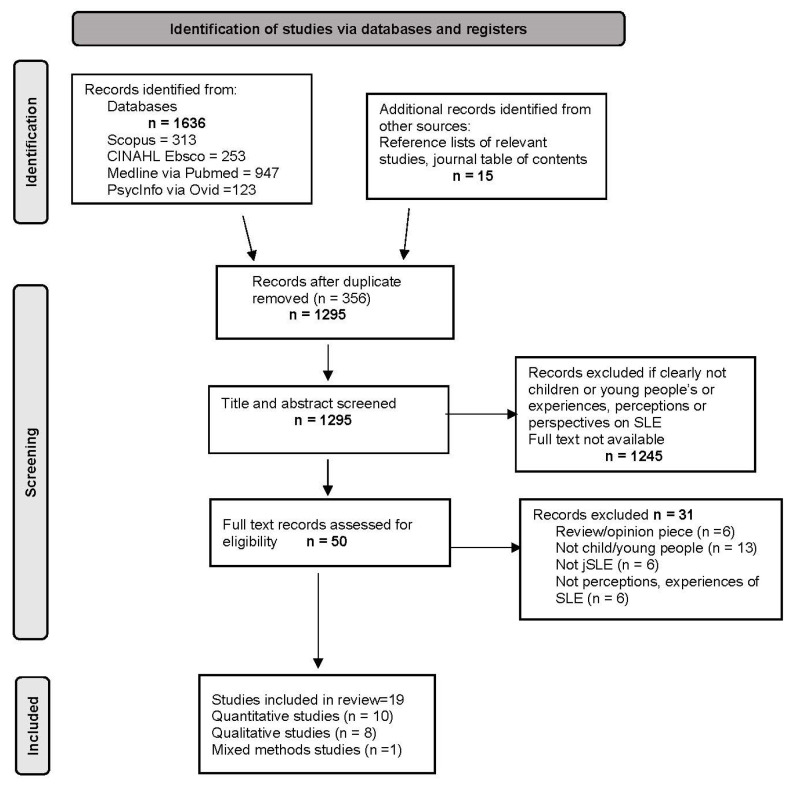
Preferred reporting items for systematic reviews and meta-analysis (PRISMA) flow chart pre-eligibility screening criteria included.

**Figure 2 children-10-01006-f002:**
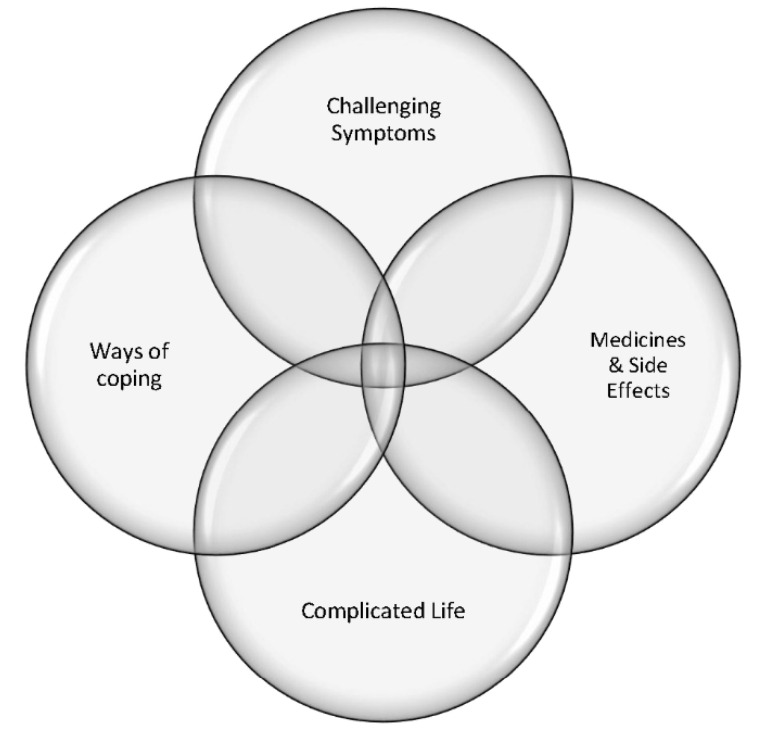
Key themes representing young people’s experience of living with cSLE.

**Table 1 children-10-01006-t001:** Systemic lupus erythematosus integrative review search terms.

Research Question: What Are the Experiences and Perspectives of Children and Young People Living with SLE/Lupus/Juvenile Onset Systemic Lupus Erythematosus?		
Population: Children and young people and youth Exposure: Living with SLE or lupus Outcomes/themes: Experiences/perceptions/perspectives		
Search terms		
Population	Exposure	Outcomes/themes
Child*	Lupus	Perception*
Young people	Systemic Lupus Erythematosus	Perspective*
Young person	SLE	Experience*
Young adult*	Juvenile onset Systemic Lupus Erythematosus	View*
Youth*	cSLE	Opinion*
Teen*	cSLE	Attitude*
Adolescent*	Exposure	Reflection*
Pediatric	Lupus	Belief*
Pediatric		Qualitative
		Phenomenology*
		Hermeneutic*

**Table 2 children-10-01006-t002:** Characteristics of included studies.

Citation	Country	Aim and Design	Sample	Data Collection	Findings	Strengths and/or Limitations
[37]	USA	Aim: To gain insight into the emotional challenges posed by lupus. Qualitative study–narrative case study.	Family groups (*n* = 3) (only 1 family group met criteria), 18 years of age from 3 clinics.	Focus groups.	Body changes and uncertainty about progressions of disease most impactful. SLE impacts on daily life and relationships. Medication compliance is challenging. Feeling of responsibility for parents’ wellbeing.	Limitations include a small sample with only 1 of the three families meeting our age criteria and poorly reported methodology, and unconventional approach to reporting. Strengths include rich data from narratives.
[38]	USA	Aim: To estimate the health-related quality of life (HRQOL) of children with childhood-onset systemic lupus erythematosus (cSLE) and compare it to that of normative cohorts; to assess the relationship of HRQOL with cSLE disease activity and damage; and to determine the effects of changes of disease activity on HRQOL. Quantitative study.	Patients with cSLE (*n* = 98) were followed every 3 months 60 Caucasian, 32 African American, 4 Asian, and 2 mixed-race patients (88 non-Hispanic, 10 Hispanic) from 7 centers.	Health-related quality of life (HRQOL) Pediatric Quality of Life Inventory Generic Core scale (PedsQL-GC), the Rheumatology Module (PedsQL-RM), the Child Health Questionnaire (CHQ), and the British Isles Lupus Activity Group Index (BILAG) was used to measure organ-system-specific disease activity. Physicians rated the course of cSLE between visits.	Active disease, the worsening of disease, and the presence of disease damage are all risk factors for poor HRQOL in cSLE. Children with cSLE have markedly lower CHQ-P50 scores than healthy children. Patients with cSLE have much lower scores on the PedsQL-GC. Control of disease is an important way to improve HRQOL in cSLE. Disease activity has a differential effect on HRQOL in cSLE, depending on the organ system involved. Lower HRQOL with the presence of Raynaud’s phenomenon.	Limitations include ratings of the change in cSLE patients’ health consistently from the patient were not collected. Thus, for the analysis, these ratings were simply combined. Most patients were adolescents, and findings, therefore, may not be applicable to younger children with cSLE. Numerous confounders of HRQOL, including socioeconomic status (SES), were not considered in the study. Strengths include a very thorough, rigorous study.
[39]	USA	Aim: To explore challenges that patients with SLE and cSLE face to identify modifiable influences and coping strategies in patient experiences. Qualitative study–phenomenology	Individuals (11–46 years) with SLE (*n* = 13), including cSLE (*n* = 7), mean age at diagnosis was 12 years, from 2 hospitals.	Focus groups.	Themes identified were challenges with SLE diagnosis and management; patient coping strategies, and modifiable factors of the SLE experience. Participants identified five primary challenges: diagnostic odyssey, public versus private face of SLE, SLE-related stresses, medication adherence, and transitioning from paediatric to adult care. Coping strategies and modifiable factors included social support, open communication about SLE, and strong patient–provider relationships. Several participants highlighted positive lessons learned through their experiences with SLE, including empathy, resilience, and self-care skills.	Strengths include the sample included individuals diagnosed with SLE at a variety of ages. Clear identification of data from the cSLE group. Findings section provides detailed analysis and is supported with appropriate quotes from participants.
[40]	Brazil	Aim: To analyze the social representations of chronic disease and its treatment from the perspective of adolescents and/or their caregivers.	Adolescents (*n* = 31), 11–21 years—4 boys and 27 girls from 1 hospital.	Free Association Words Test (FAWT), asked to associate five words to each one of the stimuli contained in the questions, “What comes to your mind when I say chronic disease?”; and then, “What comes to your mind when I say treatment of chronic disease?”.	Stimulus 1- (SLE) evoked sadness and pain (to a greater degree), medication, difficulty, learning, no cure, care, disease, limitation, faith, joy, and bad. Stimulus 2 (treatment)—medication and strength (to a greater degree), hope, improvement, consultation, discipline, joy, responsibility, health, needed, professional, help, care, cure, the word “no”, patience, bad, and affection. Differences noted between age and level of education treatment were seen positively (improvement, hope, help), with active participation of the patient (knowledge, obedience, schedule, medication).	Limitations include the use of an unconventional research methodology. Results differed between age groups and education, and this was acknowledged as a limitation. No collection of demographic characteristics which could influence social perception. No indication of disease severity and or medication use that would have influenced the response regarding disease.
[41]	USA	Aim: To estimate the prevalence of depression and medication non-adherence, describe demographic and disease characteristics associated with depression and medication non-adherence, and evaluate the correlation between depression and medication non-adherence and evaluate the association between depression and medication non-adherence in cSLE patients. Quantitative study.	cSLE (*n* = 51), 7–22 years, the subgroup analysis was limited to participants with ages between 12 to 18 years (*n* = 36) from 1 hospital.	The Patient Health Questionnaire-9 (PHQ-9) measured depression, the Medication Adherence Self-Report Inventory (MASRI) measured medication non-adherence, Systemic Lupus Erythematosus Disease Activity Index (SLEDAI).	Positive depression screen in this study population of patients with cSLE was high (59%); the high rate of reported suicidal ideation- Nearly one-quarter (23%) of participants in this study who had a positive depression screen reported suicidal ideation at the time of the screen; patients who reported medication non-adherence in this study were more likely to have longer disease duration than those who reported medication adherence; prevalence of medication non-adherence in this study population of patients with cSLE was also notable with 20% of the participants reporting less than 80% adherence.	Limitations include a small sample from one hospital. The tools used to define the exposure and outcome (PHQ-9 and MASRI, respectively) each had inherent limitations. The study has demonstrated the feasibility of administering. depression and medication non-adherence screens within a clinical setting.
[42]	USA	Aim: To assess disease characteristics, fatigue, pain, psychological symptoms and HRQoL in patients with cSLE; to identify significant predictors of reduced HRQoL at follow-up and to use the most relevant predictors to create a risk stratification (High and Low Risk) for persistently poor HRQoL of adolescents with cSLE and describe the profile of disease and psychosocial characteristics for each of these groups. Quantitative study.	Children and adolescents with cSLE, 8 to 20 years, (*n* = 60) at first appointment; (*n* = 50) children at follow-up from 1 hospital; 84% female, with 23 (46%) African American and 23 (46%) Caucasian, 11–20 years, with mean age 16.2 SD +/−2.5 years.	The Pediatric Quality of Life Inventory Multi-dimensional Fatigue Scale (PedsQL-FS); the Brief Pain Inventory (BPI); the Pain Catastrophizing Scale (PCS); Children’s Depression Inventory Version I (CDI-I); Screen for Child Anxiety Related Emotional Disorders (SCARED); Pediatric Quality Of Life Inventory; Generic Core scale 4.0 (PedsQL-GC); PedsQL-RM; Functional Disability Inventory (FDI); the SLEDAI; measures of disease activity and patient-reported measures of health-related quality of life, pain, depressive symptoms, anxiety and disability were collected at each visit.	Using clinically relevant cut-offs for fatigue and depressive symptoms, patients were assigned to Low (*n* = 27) or High Risk (*n* = 23) groups. At visits 1 and 2, respectively, clinically relevant fatigue was present in 66% and 56% of patients. Clinically significant depressive symptoms in 26% and 24%; clinically significant anxiety in 34% and 28%. Poorer health-related quality of life at follow-up was significantly predicted by higher fatigue and depressive symptoms at the initial visit.	Limitations include that most participants were adolescents recruited from 1 hospital, which may limit generalizability of the findings for younger children, and correction for multiple comparisons in the data analysis was not undertaken. This was the first study to identify potentially modifiable predictors of impaired HRQoL in cSLE children/adolescents.
[43]	USA	Aim: To characterize factors influencing self-management behaviors and quality of life in adolescent and young adult (AYA) patients with childhood- onset systemic lupus erythematosus and to identify barriers and facilitators of treatment adherence via focus groups. Mixed Methods study.	Adolescents with cSLE ages 12–17 (*n* = 10) years and young adults with cSLE ages 18–24 years (*n* = 12) from 1 hospital or from the hospital’s cSLE active clinic registry.	Patient-Reported Outcomes Measurement Information System (PROMIS) Pediatric Short Form v1.0–Fatigue; the Pain Intensity Visual Analog Scale; SLEDAI; Focus groups.	Adolescent (*n* = 10) insurance cover–public cover (*n* = 6), private cover (*n* = 4). Adolescent PROMIS fatigue score (range) 57.5 (30.3–74.4). Adolescent Child Pain VAS mean (range) 3 (0–8). Adolescent number of medications (SD) 4.6 (2.5). Young adult (*n* = 12) insurance cover–public cover (*n* = 6), private cover (*n* = 6). Young adult PROMIS fatigue score (range) 57.4 (33.1–72.4). Young adult number of medications (SD) 3.5 (1.3). Themes included knowledge deficits about cSLE, symptoms limiting daily function, specifically mood and cognition/learning, barriers, facilitators of adherence, worry about the future, symptoms limiting daily functioning, pain/fatigue, self-care and management, impact on personal relationships, and health care provider communication and relationship.	Limitations include the sample being one of convenience and from 1 hospital and being predominantly Caucasian, which does not capture the ethnic groups with the highest morbidity. Strengths include using a design that used a quantitative and qualitative approach.
[44]	Colombia	Aim: To describe how adolescents nearing transition perceive lupus. Qualitative study—grounded theory.	Adolescents with SLE ages 15–18 years (*n* = 9) (7 females and 2 males), from 1 hospital.	Semi-structured interviews.	Varied interpretation and understanding about what lupus is/does to the body. Lupus and cancer seen as having similar consequences. Sense of guilt and self-blame re: diagnosis, blaming external factors for SLE such stress or food, invisible and visible forms of lupus, and taking responsibility and action for looking after self. Healthcare staff minimized symptoms- interactions and information sharing is key to learning and taking responsibility. Life with lupus is complicated and limiting- differences from others- but creates growth and sense of responsibility.	Limitations include a small sample size, and methodological underpinnings were poorly described. Strengths include the results being clearly presented with good use of quotes.
[45]	China	Aim: To examine the relationship between physical appearance concern and psychological distress in female adolescent patients with systemic lupus erythematosus. Quantitative study.	Female adolescents with SLE (*n* = 84), mean 15.3 SD +/−1.4 years of age from 1 hospital.	Appearance concern was assessed by the physical appearance domain of the multi-dimensional Self Perception Profile for Children (SPPC). The SLEDAI was used to assess disease activity. Depression–the CDI covers negative mood, interpersonal difficulties, negative self-esteem, ineffectiveness, and inability to experience pleasure.	CDI: Nearly all patients showed increased depressive symptoms, as indicated by the mean scores on the CDI. The total mean CDI was 18.5 ± 4.3, indicating that these patients were experiencing depressive symptoms. Furthermore, a total of 32 (38.1 %) patients had CDI larger than 19 points. Among the CDI subscales, negative mood, negative self-esteem, and anhedonia contributed to the total scores mostly. Physical Appearance: Regarding the concern about appearance, 77 (91.7 %) patients with SLE reported that they felt unattractive due to the disease, according to the questions from the SPPC. Among the SLE patients, the SPPC physical appearance score was 13 ± 2.8, indicating that the SLE adoles cents believe their appearances were severely impaired. Appearance concern was highly correlated with depression (*r* 0.758, *p* 0.001). Subsequently, age was moderately correlated with depression (*r* 0.468, *p* 0.001). Other variables, such as the number of admissions and disease duration, were partially correlated with depression.	Limitations include the use of a generic measure for physical appearance evaluation and is less robust than a more comprehensive one and less effective than a SLE-specific evaluation tool. The self-administered evaluation of depression through the CDI could have failed to capture the whole depressive disorders. The authors failed to evaluate appearance concerns and psychological conditions of the adolescents before the SLE onset because the patients who received these questionnaires had been treated with an average of 15.8 months, and male adolescents were excluded.
[46]	USA	Aim: To evaluate pain, fatigue, and psychological functioning of childhood-onset systemic lupus erythematosus (SLE) patients and examine how these factors impact health-related quality of life (HRQOL). Quantitative study.	Children and adolescents with cSLE (*n* = 60), 8–20 years, mean 16.1 SD +/−2.5, years of age from 1 hospital.	The visual analog scale (VAS) of pain intensity (0–10), the Adolescent Sleep Wake Scale (ASWS), FDI, the PedsQL–FS, Pain Coping Questionnaire (PCQ), the PCS, SLEDAI, the CDI-I, the SCARED questionnaire, and the PedsQL–GC scale and PedsQL–RM module.	The PedsQL-GC, -RM, and -FS summary scores were significantly lower in the childhood-onset systemic lupus erythematosus population than in reference populations of healthy children, and those with arthritis. Although most of the childhood- onset SLE patients reported no more than minimal functional disability, 18% (11 of 60) reported moderate to high functional disability (FDI > 13). Of the childhood-onset SLE patients in the study sample, 65% (39 of 60) reported to have fatigue on the clinician-administered checklist, with 40% (24 of 60) reporting clinically relevant pain (pain-VAS of 3). Further, 30% (18 of 60) of the study participants reported clinically important depressive symptoms (CDI-I of 12), 37% (22 of 60) reported clinically relevant anxiety, and 22% (13 of 60) reported a high level of pain catastrophizing. On average, fatigue, anxiety, and depression were not found to be significantly greater among patients on steroid therapy compared to those not requiring steroid therapy (fatigue [PedsQL-FS] score: 55.0 versus 60.5, anxiety [SCARED] score: 24.1 versus 20.7, and depression [CDI-I] score: 9.6 versus 9.8; all *p* values were not significant). The scores of the fatigue (PedsQL-FS), anxiety (SCARED), and depressed mood (CDI-I) measures were all highly and sig nificantly correlated with HRQOL (PedsQL-GC and -RM). The BPI and pain-VAS were highly correlated and also featured similar correlations to the remaining variables. only. HRQOL measured by PedsQL-GC was significantly impacted by fatigue and pain (R^2^ 5 0.75, *p*, 0.001), with fatigue predicting 42% and pain predicting 33% of the model variance, but pain coping, anxiety, pain catastrophizing, and depression did not significantly predict PedsQL-GC scores. The PedsQL-RM was significantly impacted by pain, fatigue, and anxiety (R^2^ 5 0.71, *p*, 0.001), with pain predicting 33%, fatigue predicting 25%, and anxiety predicting 7% of model variance, but pain coping, pain catastrophizing, and depression did not significantly predict PedsQL-RM scores. Both support the notion that pain, fatigue, and to a lesser extent anxiety account for significantly diminished HRQOL observed in childhood-onset systemic lupus erythematosus patients.	Limitations include the majority of the patients being teenagers, the sample size being small, and the design being cross-sectional. Strengths include the patients studied were well-phenotyped and are representative of those followed at the hospital.
[47]	USA	Aim: To assess emotional and behavioral problems in children and adolescents with SLE during the remission of disease activity. Quantitative study.	Cohort 1: children and adolescents with cSLE disease (*n* = 38), 10–18 years old from 1 clinic. Cohort 2: young adults with cSLE, (*n* = 16), 18–24 years from 1 clinic.	Cohort 1: CDI, Intelligence test, Pediatric Quality of Life Index, SLEDAI, Systemic Lupus International Collaborating Clinics Damage Index (SDI). Cohort 2: Intelligence test, Beck Depression Inventory (BDI-II), Health Survey, SLEDAI, Systemic Lupus International Collaborating Clinics Damage Index (SDI).	In cohort 1 10 patients had elevated depression scores (26%), and three (8%) subjects had clinically significant depression scores (*T* score 65). No subject endorsed suicidality. In cohort 2, seven (44%) of the young adults had elevated depression scores (10), with two (12.5%) subjects scoring in the moderate to severe depressive symptoms range (>19). Similarly, no subject endorsed suicidality. Symptoms receiving the highest mean ratings by cohort one included fatigue, school problems, indecisiveness, despair, and sleep disturbances. The most severe individual depressive symptoms endorsed by cohort two included sleep disturbances, loss of energy, fatigue, changes in appetite, and indecisiveness.	Limitations include a small sample size, allowing for hypothesis generation and potentially limiting our ability to detect differences even if they were present and data collection was cross-sectional.
[48]	USA	Aim: To determine the relationship between HRQOL, disease activity, and damage in a large prospective international cohort of cSLE. Quantitative study.	Children and adolescents with SLE (*n* = 456), (384 females) from 39 centres across four continents (North America, South America, Europe, Asia).	SMILEY, PedsQL–GC scale and PedsQL–RM module, SLEDAI, Physician’s Global Assessment (PGA), Systemic Lupus International Collaborating Clinics/American College of Rheumatology Damage Index (SDI)–data collected across 6 collection points V1-V6).	The highest domain scores were: Social and Physical (PedsQL Generic), Daily Activities (PedsQL Rheumatology), and Social (SMILEY). Lowest domain scores were: Emotional (PedsQL Generic), Worry (PedsQL Rheumatology), Effect on Self, and Burden of SLE (SMILEY). Patients with higher disease activity (SLEDAI > 12, PGA > 2), higher damage (SDI > 2), and current/past cyclophosphamide and/or rituximab use had lower SMILEY Limitation domain scores at visits V1 (*p* < 0.05 for all values for SMILEY). Patients with higher disease duration did not have lower SMILEY Limitation domain scores except at V3. PGA, SLEDAI and SDI, and SMILEY scores did not differ between the two genders; however, child SMILEY scores were lower in girls.	Limitations include missed information on children who were excluded or the reason for attrition. The strengths include a large sample across four continents and the use of validated tools.
[49]	USA	Aim: to explore children’s (and adolescents’) perception of the impact of SLE on school, the relationship between child and parent reports on school-related issues, and the relationship between health-related quality of life (HRQOL) and school-related issues. Quantitative study.	Children and adolescents with SLE (*n* = 41) (73% girls), 9–18 years, mean age was 15 SD +/−3 years from two centers.	SLE-specific HRQOL scale, SLEDAI, Simple Measure of Impact of Lupus Erythematosus in Youngsters (SMILEY), PedsQL–GC scale, and PedsQL–RM module.	Mean school domain scores for children of the PedsQLTM generic report were lower compared with total and subscale scores. Patients reported difficulty with schoolwork, had problems with memory and concentration, and were sad about the effect of SLE on schoolwork and attendance. Eighty-three percent of patients felt that they would have done better in school if they did not have SLE. Moderate correlations (r = 0.3–0.4) were found between SMILEY_ total score and the following items: satisfaction with school performance, interest in schoolwork, remembering what was learned, and concentrating in class. Patients on intravenous chemotherapeutic medications missed more school days (*p* < 0.05) compared with patients on oral medications.	Limitations include potential recall bias among patients and the need for a larger sample so children’s responses could be stratified by age. The strengths include data collected across two centers and the use of validated tools.
[50]	USA	Aim: To identify domains that are critical in determining QOL in children with SLE. Qualitative study–grounded theory.	Children with SLE (*n* = 38), (30 females) 6–20 years from 1 center.	Interviews and written responses.	Limitations of SLE are physical, social, and psychological. Impact on daily life, on social and family relationships. Impact on self- related to medication side effects and disease effects. Sadness about diagnosis-rationalizing and normalizing as coping mechanisms. Stress and anxiety about the future.	Limitations include a small sample, poorly described methodology, and data were collected from 1 center.
[51]	Italy	Aim: To assess the health-related quality of life (HRQL) of patients with juvenile-onset systemic lupus erythematosus (CSLE) and its relationship with disease activity and accumulated damage. Quantitative study.	Children and adolescents with juvenile-onset systemic lupus erythematosus (CSLE), (n= 297), (252 female), mean 16.2 SD +/−4.9 years of age from 8 hospitals and groups across four countries (Europe, the US, Mexico, and Japan).	CHQ, SLEDAI, SDI	Most impaired CHQ subscales were global health, general health perceptions, and parent-impact emotional. Compared with healthy children, CSLE patients had lower values in all subscales of the CHQ. The progressive decline of HRQL with increasing disease activity and accumulated damage effect more pronounced in both instances on the physical than in the psychosocial health domain. Greater impairment of HRQL occurred in patients with active disease in the central nervous, renal, and musculoskeletal systems. Active nephritis and seizure most significantly affected family life (PE, PT, and FA). Lupus headache was the only disease manifestation that impaired Mental health. Pleurisy and fever had a more significant impact, respectively, on BP and CH. Marked reduction of self-esteem associated with renal disease. HRQL abnormalities were more associated with clinical features than with laboratory abnormalities.	Limitations include the use of a cross-sectional design where it is difficult to determine a causal relationship since there is doubt about the timing of when a child is in remission or in the middle of an exacerbation of SLE.
[52]	United Kingdom	Aim: To explore in depth the views of CSLE patients and parents on potential treatment targets (e.g., LLDAS), outcome measures (e.g., HRQOL and fatigue measures), and study designs being considered by TARGET LUPUS in light of their previous treatment and care. Qualitative study.	Children and adolescents with CSLE, (*n*= 12), (10 female), 9–18 years, mean age 14 years from eight centers.	Semi-structured interviews.	Variation in treatment response. Symptoms that represented lupus low disease activity state (LLAS) were variable and different from adults. Intolerable visible signs of lupus- steroids are undesirable-symptoms–medicine side effects- fatigue is a significant symptom—wanted to minimize disruption from CSLE in all aspects of the patient’s life and preferred treatment goals to include corticosteroid dose reduction, HRQOL, and fatigue in addition to the targeting of disease activity.	Strengths include the methods section being clear and concise. Good inclusion of quotes from participants and data collected from eight centers.
[53]	Singapore	Aim: To explore experiences in medication adherence among adolescents with SLE. Qualitative study.	Adolescents with SLE (*n* = 14), 11–19 years, mean age 15.4 SD 2.06 years from one hospital.	Semi-structured interviews.	Adjusting and creating a new normal that included medicines- contending with the side effects of medications was challenging- understanding the ‘why’ encouraged adherence. Participants felt that viewing a graphical model of their blood test results over their disease courses provided clearer representations of how medications or the lack of such influenced their conditions over time. Avoiding hospitalization and being sick were also goals that the participants strived toward, further increasing their medication-taking motivations. Participants resented doctors’ lack of transparency when they explained the medications’ side effects. Taking steroids was of the greatest concern because these medications made them gain weight.	Limitations include inconsistencies with reporting participant numbers. Strengths include a methodologically sound study that was well described.
[54]	Australia	Aim: To describe the experiences, perspectives, and health care needs of adolescents and young adults diagnosed with SLE prior to age 18 years. Qualitative study–grounded theory.	Adolescents and young adults (*n* = 26), (24 female), 14–26 years, mean age 18 years from five hospitals.	Focus groups and interviews.	Being treated differently, reluctance to disclose. Poor self-image felt different from peers. Physical manifestations of disease and medication side effects impacted negatively on the image of self. Isolation and stigma. Symptoms impacted future aspirations (jobs, parenthood, study). Lack of age-appropriate information- uncertainty about the future- Knowledge of SLE was mixed. Desire for autonomy and developing self-reliance. Positive side to SLE: greater confidence and strength of character. Successful management relied on family and friends. Relationship with clinicians- important to have trusted relationships- some talked about being ‘judged’.	Limitations include some methodological confusion in the description of the analysis approach undertaken.
[36]	Turkey	Aim: To assess the peer victimization, depression, anxiety, self-esteem, and QOL levels and compare them with those of the control group; to evaluate the association between QOL, psychological symptoms, and peer victimization; and to examine the determinants of QOL in these patients. Quantitative study.	Children (*n* = 9) and adolescents (*n* = 32) with SLE, 32 females), 9–18 years, mean 14.7 SD +/−2.6 from 1 hospital.	SLEDAI, Peer victimization scale (PVS), CDI, State-trait anxiety inventories for children (STAIc), PedsQL, Rosenberg’s self-esteem scale (RSS).	Peer victimization, self-esteem, depression, anxiety, and QOL levels of the patients with SLE were not worsened than those of the control group. Peer victimization and trait anxiety levels have roles in determining the QOL impairment of children and adolescents with SLE. The difference in levels of peer victimization between the patients and controls was not statistically significant. SLE patients with lower disease activity may have lower depression and anxiety levels compared with those with higher disease activity.	Limitations include data collection being from 1 hospital, a structured psychiatric interview was not performed, and a disease-specific questionnaire for measuring QOL was not used.

Terms and Tools: ASWS Adolescent Sleep Wake Scale, AYA Adolescent and Young Adult, BDI-II Beck Depression Inventory, BILAG British Isles Lupus Activity Group Index, BPI Brief Pain Inventory; CHQ Child Health Questionnaire, cSLE childhood-onset systemic lupus erythematosus, CDI -I Children’s Depression Inventory Version I; Free Association Words Test (FAWT), FDI Functional Disability Inventory; HRQOL health-related quality of life, CSLE juvenile-onset systemic lupus erythematosus, LLAS Lupus Low Disease Activity State, MASRI Medication Adherence Self-Report Inventory, MASC Multi-dimensional Anxiety Scale for Children, PCS Pain Catastrophizing Scale; PCQ Pain Coping Questionnaire, PHQ-9 Patient Health Questionnaire-9, PedsQL-GC Pediatric Quality of Life Inventory Generic Core scale, PedsQL-FS Pediatric Quality of Life Inventory Multi-dimensional Fatigue Scale; PedsQL-RM Rheumatology Module, PVS Peer victimization scale, PROMIS Patient-Reported Outcomes Measurement Information System, PGA Physician’s Global Assessment, RSS Rosenberg’s self-esteem scale, SCARED Screen for Child Anxiety Related Emotional Disorders; SPPC Self-Perception Profile for Children, SMILEY Simple Measure of Impact of Lupus Erythematosus in Youngsters, STAIc State-trait anxiety inventories for children, SDI Systemic Lupus International Collaborating Clinics Damage Index, SLE systemic lupus erythematosus, SLEDAI Systemic Lupus Erythematosus Disease Activity Index, SDI Systemic Lupus International Collaborating Clinics/Damage Index.

**Table 3 children-10-01006-t003:** Illustrative Quotes (place holder).

Illustrative Quotation Reflecting Each Theme		
Participants Quotations and/or Authors Explanations		Contributing References
Challenging Symptoms Disruption to life and altered self	I can’t go outside to play with my own friends. I lost my old friends. I can’t do the stuff that kids of my age do. Cause sometimes I feel tired and boring. Cause when I got lupus, I change cause now I can’t do the stuff I could do before. Everything in my life changed [44] Feeling isolated, even though you are with the people you care about, you just still feel isolated, because you can’t do what they are doing [50] I don’t really tell all of my friends. I was actually ashamed to tell anyone at first when I was diagnosed [39]	[37,39,43,44,45,50,52,54]
Severity	I believe that there are several types of lupus, but mine is like… I don’t know… tougher [44]	[38,39,50,52,54]
Depression and anxiety	The prevalence of a positive depression screen in this study population of patients with cSLE was high (59%) * [41] Nearly one-quarter (23%) of participants in this study who had a positive depression screen reported suicidal ideation at the time of the screen * [41]	[41,45,46,47,48]
Fatigue	Fatigue is the most difficult part of having lupus. Lupus is an evil disease that makes you sleep a lot. [43] The fatigue I feel isn’t like the flu, it’s like I’ve been knocked over by a bus [37] Fatigue, joint symptoms, and headaches had a markedly detrimental effect on the HRQOL of children with cSLE * [52]	[37,38,41,43,46,52]
Medicines and Side Effects Dreaded steroids	It was really hard getting to school. I was on prednisone and got pretty fat, so I was getting bullied a lot. It was hard [54] It was just so defeating. No matter how hard I try now, I can’t take off weight and I have this fat face. Who’s going to want to date me? [37] I just wanted to come off them. Even when I was only on half a tablet, I didn’t feel happy with being on them [52]	[39,44,52,53,54]
Conflicting feelings	All the medicine I have to take, I don’t see results right then and there... What’s the purpose of taking it? I’m going to feel the same regardless. (Young adult) [43] …Having lupus doesn’t make me feel happy. But it doesn’t make me feel sad. I wouldn’t say how lonely do you feel because of lupus. Like, I’m not sad, I’m not lonely, but I’m not happy [52]	[40,43,49,50,52,53]
Medication adherence	So sometimes if I’m busy one day and I just forget. If I forget one dose, I just start forgetting it for a couple of days [39] The prevalence of medication non-adherence in this study population of patients with cSLE was also notable with 20% of the participants reporting less than 80% adherence with SLE medications * [41]	[37,39,41,43,44,52,53,54]
Complicated Life School, sports and social activities	I’d like to go back to school and pursue my education, but I just don’t know if it’s realistic. I keep trying to get it under control, but I don’t know if there’s such a thing [37] Being sick makes others impose limitations on me [44] I couldn’t be like the rest of them [44]	[37,44,49,50]
Giving things up	You can’t be a kid in the moment, you have to think weeks and months...I know if I do this, I can’t do this the next night. And a lot of kids don’t have to do that [43] Having lupus prevents you from doing things you like a lot [44] For me lupus means sacrifices. I can’t actually do what I want to [54]	[43,44,50,54]
Quality of life	Living with lupus is complicated because of the medicines and the medical appointments [44] Patients with jSLE have poorer HRQL as compared with healthy controls in both physical and psychosocial domains, with physical health being more affected [51]	[36,38,42,44,46,48,50,51]
Lack of understanding	My teacher decided to tell all of them I had lupus, and they thought they could catch it. So, they wiped the seats off with cleaning wipes... trying not to catch the lupus [43] I was told I had lupus and my mother… well she didn’t want to tell me anything [44] When I was younger, I could have had someone explain it to me… that it wasn’t something that wasn’t going to go away [54]	[39,43,44,50,54]
Ways of Coping Family and friends	This is a great comfort to me. Talking to other SLE patients has been so helpful. Only they know what it’s like to have the disease, to feel too tired to get out of bed, to feel pain in your joints [37] What helped me the most, I had a really big group of friends, and they didn’t really care what I looked like [54]	[37,39,40,53,54]
Relationships with health providers	It’s really all about the connection with your doctor… I think it’s just the relationship that you have with your doctor [39] I will still just take medicine cos I mean it’s prescribed by the doctor…I trust that my doctor would know some of the side effects…make informed decision on dosage [54]	[37,39,40,50,52,53,54]
Maintaining positivity	I try to be optimistic and hope that if I make plans, I’ll be healthy enough when that day comes. If I’m not, I hope others will understand [37] Having lupus helps you in many ways: makes you more responsible [44]	[37,44,54]

* Authors explanations.

## Data Availability

Data available in a publicly accessible repository.

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
