# Peer review of "Experiences and Perspectives of Children and Young People Living with Childhood-Onset Systemic Lupus Erythematosus—An Integrative Review"

_children, 2023, doi:10.3390/children10061006_

Round 1

Reviewer 1 Report

Thanks for the opportunity to review this paper. This study was investigated “Experiences and perspectives of children and young people living with Juvenile onset Systemic Lupus Erythematosus. An integrative review”. This manuscript is well written, organized and interesting study. The study adds a new perspective to the relationship between living with SLE. This manuscript should be reorganized according to recommended revisions.

Major revisions:

-       Introduction section should be elaborated.  The objective of this study should be clearly explained. 

-       Material-methods section should be detailed. 

-       Results section should be specified.  This section is quite complicated, it should be simplified. 

-       Statistical methods should be checked especially parametric and nonparametric values. Percentage and numerical values should be given consecutively

-       Discussion and conclusion section should be specified. Conclusion part of discussion should be added. 

-       Diagnostic limitations of this study should be summarized in discussion section. 

Minor revisions should be done in this study.  

-        References section should be updated. References section should be written according to the rules. 

-        All abbreviations should be summarized in this manuscript.  

-        The keywords should be added by the alphabetical order

-        There are numerous spelling mistakes in the manuscript and reference section. These must be corrected. 

-        Table and figure headings should be more informative.

-        Manuscript should be reorganized according to journal rules. 

Should be improved. 

Author Response

Please find response in attached word document

Reviewer 2 Report

The authors undertook an analysis of bibliographic data on the impact of c-SLE on the psychosocial, emotional, and physical development of children and adolescents with SLE. It is clear, that children and adolescents with c-SLE experience a wide range of symptoms, such as pain, fatigue, rash, depression, anxiety and altered body image, which interfere with their normal psychosocial, emotional, physical, and developmental outcomes.

The authors selected eleven studies on the topic as above conducted in nine countries: United States of America, Brazil, Colombia, China, Italy, United Kingdom, Singapore, Australia, and Turkey. The authors identified four themes and fourteen sub-themes: 1) difficult symptoms (disrupted life and altered self, aggravation, fatigue, depression and anxiety), 2) medications and side effects (scary steroids, conflicting feelings and medication adherence), 3) difficult life (school and social sports, giving up things, lack of understanding and quality of life), and 4) ways to cope (family and friends, relationships with health care providers, maintaining a positive attitude). The authors emphasize that social support systems, including peers, family, and support groups for c-SLE patients have proven to be essential and important in coping with c-SLE. Organized events such as camps for children and adolescents with c-SLE were beneficial in providing social support through shared experiences and helped young people normalize their lives. This is not a new proposal and recommendation, known to other groups of children and adolescents with chronic illnesses such as those with cancer, diabetes, hemophilia. Authors are asked to post such a comment.

Although c-SLE shares many similarities with a-SLE, there are fundamental differences in the impact of symptoms and adverse drug reactions among children and adolescents. The review also highlighted that the significant anxiety, fatigue and depression associated with c-SLE are poorly studied and require further research to well define needs, exploration, policy and knowledge in this patient group. And this observation, according to the authors, is key, although they do not report analogous data in a-SLE patients. The authors are invited to supplement with data from papers addressing similar topics in a group of adults with a-SLE.

Minor remark:

The spelling a-SLA, cSLE should be standardized with or without a hyphen.

Author Response

Please find attached word document with response to reviewer 2

Round 2

Reviewer 1 Report

All corrections and revisons are appropriate for me

Good